# Progress in Personalized Psychiatric Therapy with the Example of Using Intranasal Oxytocin in PTSD Treatment

**DOI:** 10.3390/jpm12071067

**Published:** 2022-06-29

**Authors:** Sandra Szafoni, Magdalena Piegza

**Affiliations:** Department of Psychiatry, Faculty of Medical Sciences in Zabrze, Medical University of Silesia in Katowice, 42-612 Tarnowskie Góry, Poland; mpiegza@sum.edu.pl

**Keywords:** PTSD, personalized medicine, individualized treatment, intranasal oxytocin, psychotherapy

## Abstract

Post-traumatic stress disorder (PTSD) is a severe mental disorder that results in the frequent coexistence of other diseases, lowers patients’ quality of life, and has a high annual cost of treatment. However, despite the variety of therapeutic approaches that exist, some patients still do not achieve the desired results. In addition, we may soon face an increase in the number of new PTSD cases because of the current global situation—both the COVID-19 pandemic and the ongoing armed conflicts. Hence, in recent years, many publications have sought a new, more personalized treatment approach. One such approach is the administration of intranasal oxytocin (INOXT), which, due to its pleiotropic effects, seems to be a promising therapeutic option. However, the current findings suggest that it might only be helpful for a limited, strictly selected group of patients.

## 1. Post-Traumatic Stress Disorder—The Problem of Our Time Demanding Personalized Solutions?

The symptoms of post-traumatic stress disorder (PTSD) have been well known to humankind since ancient times. Both medical records and literary works testify to this fact [1,2]. Today, however, we know more about this disease, and we know that it can affect any person who has experienced or witnessed a direct threat to life or health [3]. This also includes patients who experienced acute coronavirus disease 2019 (COVID-19) and healthcare workers who worked during the pandemic period [4,5]. Medical professionals can develop symptoms of PTSD because of periodical exposure to traumatic ethical and moral difficulties [6]. Another factor contributing to an increase in PTSD prevalence is the ongoing military conflicts. This effects not only direct fighters but also war refugees who find shelter in various European countries. A further crucial point is the financial cost of treating this disorder. The data gathered from 2002 to 2011 in the U.S. show that PTSD as a primary diagnosis resulted in hospitalization costs of USD 1.2 billion. Simultaneously, all PTSD-related hospitalization treatment expenses were approximately USD 35 billion [7]. The overall treatment cost also arises from comorbidities. It is noteworthy that the PTSD pathophysiology could presumably contribute to the range of somatic and psychiatric diseases faced by this patient population [8,9,10,11,12,13,14]. Therefore, its appropriate prevention and treatment could be economically beneficial.

Currently, different therapeutic options exist, including pharmacological and psychotherapeutic ones. However, not all patients achieve the expected results [15,16]. For this reason, the literature has proposed various novel solutions in recent years. One such solution is intranasal oxytocin (INOXT) administration [17,18].

In this review, we show that the dynamic interplay of oxytocin’s properties might contribute to achieving good outcomes in patients suffering from PTSD. We also look at potential factors influencing different treatment outcomes, thus striving to identify the necessary points for further research. Finally, referring to the example of oxytocin (OXT), we focus on solutions that can change our view of diagnostic and therapeutic issues.

## 2. Oxytocin in Medicine—How Did it Start and Where Are We Now?

Oxytocin was discovered in 1906 when Dale noticed that a human’s posterior pituitary extract was capable of causing uterine contractions in a pregnant female cat [19]. This substance, along with vasopressin, is initially produced in the paraventricular nucleus (PVN) and the supraoptic nucleus (SON) of the hypothalamus, and it is then transported and stored in the posterior pituitary gland. Due to its properties, its intravenous form has been in use in labor induction since the 1950s [20]. As time passed, findings of other oxytocin effects began to be reported. These findings concerned its impact on social behavior, human emotionality, empathy, and attachment [21,22]. Subsequent studies also indicated the anxiolytic and antidepressant nature of this neuropeptide [23]. Moreover, animal findings have shown the wide brain distribution of oxytocin receptors, maintaining the hopes of its potential treatment benefits. These localizations were the amygdala, hippocampus, and prefrontal cortex [24,25,26]. The most popular form of oxytocin administration today is oxytocin in an intravenous form due to its role in obstetrics. However, the size of the substance makes it impossible for it to effectively cross the blood–brain barrier, consequently preventing the achievement of the desired concentration in the brain and, thus, preventing the induction of specific effects. To overcome this difficulty, initial studies used intranasal oxytocin (INOXT) augmentation. Via the blood vessels in the nasal cavity, the substance can directly enter the cerebral circulation, having 2% bioavailability. Recent studies have shown that more than 95% of the oxytocin available in the brain is transported through the nasal cavity [27,28]. Currently, studies using this form of administration are examining the potential of oxytocin in the treatment of various mental disorders. Studies have already been carried out on the effect of oxytocin on patients with schizophrenia, drug and alcohol use disorder, autism spectrum disorder, and borderline personality disorder [29,30,31,32]. The same applies to post-traumatic stress disorder. In our opinion, the most promising is using OXT in enhancing prolonged exposure (PE) therapy despite the lack of statistically significant findings on symptom improvement [33]. However, there are also other types of studies where OXT was administered to prevent PTSD development. Those studies also did not point out a significant group difference in the Clinical Administered PTSD Scale (CAPS) total score, but that was reported in participants with high baseline CAPS scores [34]. Therefore, this indicates that oxytocin could be of worth to a specific group of patients with PTSD.

In this review, we attempt to identify the main factors responsible for different therapeutic responses. Moreover, we focus on the OXT effects that may be crucial given the pathophysiology of the disorder. However, we do not only analyze reports from studies in subjects with PTSD, as too-little research has been conducted to date.

## 3. Pleiotropic Effects of Oxytocin

Oxytocin is most frequently linked to empathy and attachment to another person. Recent studies, however, point to its more widespread, pleiotropic effects. These effects appear to be particularly essential within the context of PTSD pathophysiology. They might result directly from interaction with several brain structures and probably neurotransmitter systems, such as the serotonergic and GABAergic systems [35,36,37,38]. The oxytocinergic system encompasses receptors and key areas of brain projections in terms of both the etiology and progression of PTSD symptoms (Figure 1).

Specifically, we mean the amygdala, hippocampus, ventromedial prefrontal, and anterior cingulate cortex [24,25,28]. Also noteworthy is the presence of OXT receptors in the raphe nuclei—clusters of serotonergic neurons located in the brainstem and CRH neurons located in the hypothalamus [35,39]. Perhaps it is for this reason that the administration of oxytocin has been found to result in a reduction in anxiety and depressive symptoms, bearing in mind their role in the etiology of these disorders [40]. This, along with an increased attachment and trust in a therapist, can reduce the barrier to engaging in the psychotherapy process. It is also associated with avoidance behavior reduction [41]. Avoidance is one of the forms of coping with a stressful stimulus. It is established quickly due to the reduction in a patient’s exposure to an unpleasant stimulation, which, in this case, manifests as perceived fear. Thus, that behavior hampers the confrontation of the traumatic memory and, consequently, prevents the proper functioning of fear extinction memory. In general, extinction memory consists of creating a new memory path. This path combines the stimulus that triggers the fear reaction (e.g., traumatic event memory) with a new, safe stimulus from the patient’s point of view. Furthermore, it should be stressed that the creation of a new memory pathway does not automatically change someone’s way of reacting to the memory. However, if the safe exposure to the stimulus is repeated many times, the new association will successively dominate through classical Pavlovian fear conditioning. This mechanism is widely known and used today in exposure therapy [42,43].

Regarding OXT effects on extinction memory, human and animal studies are available. According to their findings, this neuropeptide influence is mainly due to the GABAergic neurons in the amygdala basolateral complex, central amygdala, and infralimbic areas of the medial prefrontal cortex [36]. However, human studies show 121 mixed results, and as it occurs, the response might depend on the phase of extinction 122 memory in which the substance was administered [36]. To conceive oxytocin’s therapeutic potential, it is indispensable to outline the stress physiology and other aspects of PTSD pathophysiology. Generally, different stressors are initially recognized and interpreted by the amygdala. This structure sends its projections to the prefrontal cortex and the hippocampus, among others, which allows for the body’s responses to the stimulus to be coordinated. Interestingly, according to the latest research, the intranasal administration of oxytocin can modulate the reactivity of this structure. For instance, one study involving healthy men showed increased connectivity between the amygdala and the hippocampus [44]. Furthermore, according to a meta-analysis of 66 fMRI studies, the intranasal administration of oxytocin decreased amygdala activity in both healthy and psychiatric populations [45]. The direct influence of oxytocin probably occurs in the hypothalamus at the level of CRH neurons [46]. It may result from oxytocin binding to the OXT receptors located on these neurons. Consequently, it likely leads to the inhibition of CRH secretion, cascadingly affecting subsequent components of the HPA axis (Figure 2).

This might be relevant in cases with deviations in CRH and cortisol levels, which have been observed in the PTSD population. The data show that CRH levels increase in cerebrospinal fluid [47,48]. This may indicate the frequent activation of the stress response, which would be consistent with the up-to-date knowledge of the disorder’s pathophysiology. However, in the matter of cortisol blood concentration, the results remain divergent. In some patients, the blood concentration of cortisol was found to be within physiological norms, while in others, it remained increased or decreased [49,50,51]. Of course, many factors could have caused this, such as the general condition, disorder duration, the severity of symptoms, and the existing compensation mechanisms. Nevertheless, knowing the numerous functions and properties of cortisol, and its general impact on the stress response, one of the goals of therapy should be to minimize the re-experiencing of traumatic memories and, therefore, achieve proper hormone concentrations. We could thus obtain as a potential benefit the lack of development or the inhibition of the progression of some co-occurring diseases [52], such as metabolic, autoimmune, and cardiovascular disorders [8,9,10,11]. Metabolic disorders, such as insulin resistance or its decompensated form, type 2 diabetes, might, to some extent, result from the action of cortisol and the frequent activation of the sympathetic nervous system [52,53]. This stems from the fact that the activation of the HPA axis and sympathetic nervous system leads to inflammatory reactions [54,55]. In part, this has been substantiated by studies on the PTSD patient population, in whom significantly increased levels of pro-inflammatory cytokines, such as Il-1B, Il-6, TNF-a, and IFN-γ, were observed [55,56,57,58]. Intriguingly, some papers report their roles in the appearance and development of neurodegenerative diseases. Such an effect may arise from the release of pro-inflammatory cytokines from the microglia, the inhibition of neurogenesis, and the promotion of nervous apoptosis [59,60]. Over time, this can lead to morphological and functional changes in different brain regions. Recent evidence suggests that the dysregulation of the immune response may lead to impaired memory extinction [55]. In this context, a study conducted on an animal model with OXT seems interesting. The results indicated a potential decrease in IL-1B and TNF-a levels after the administration of OXT, which was used as a substance to enhance memory extinction [61]. The inflammation that occurs in the PTSD population, as already stated, results partially from the increased activity of the sympathetic system. In these patients, there is a dysregulation consisting of the predominance of sympathetic activity over parasympathetic activity. However, according to the latest research, oxytocin may also prove helpful in this respect. This stems from the fact that OXT affects upregulation or increases the insensitivity of alpha-2 adrenoreceptors [62]. Moreover, one study reported an increase in vagal activity after oxytocin administration. This nerve activity demonstrates parasympathetic system reactivity [63]. Notably, decreasing the sympathetic tone with oxytocin could result in a reduction in co-occurring sleep disturbances [64]. Sleep, in which we can distinguish two phases, namely, rapid eye movement (REM) and NREM (non-REM), plays a crucial role in memory pathway consolidation and emotional regulation. The REM phase seems to be more crucial from the standpoint of PTSD [65,66].

This derives from the frequent observations of sleep fragmentation and the co-occurrence of nightmares during that phase [67]. The occurrence of nightmares in REM sleep is related to an increased level of norepinephrine, resulting from the decreased activity of the parasympathetic system [64]. Therefore, a patient’s emotional response and anxiety related to existing nightmares must not diminish. Importantly, the long-term memory pathways for emotional memories are stabilized during REM sleep. Hence, not only would minimizing sleep disturbances lead to an improvement in a patient’s quality of life, but it could also significantly contribute to better treatment outcomes [65] (Figure 3).

## 4. Oxytocinergic System in Light of Inter-Individual Variable Factors

Today, it is well known how crucial inter-individual factors in terms of treatment response can be. Everything points to the fact that the same is true for oxytocin treatment. What we have here is the overlapping of both non-modifiable and modifiable factors. The non-modifiable group includes the patient’s age or sex, whereas the modifiable ones might include the blood concentration of some steroid hormones [68]. Regarding non-modifiable factors, older age is evidently associated with metabolic changes and a more frequent coexistence of other diseases. Hence, a patient’s gender is relevant to hormonal profiles and the functioning of several crucial brain regions.

In particular, sex differences are marked in social behavior, fear conditioning, and sleep architecture, that is, in all the key aspects of PTSD [69]. In this context, one of the most important hormones is estrogen, being an extracellular regulator in some animals [25]. Indeed, we still do not know how this substance influences the oxytocinergic system in humans, but it seems to play a crucial role. Indirect evidence at this point would pose that a higher oxytocin blood concentration would be obtained in women than in men [18]. Nevertheless, returning to the issue of social functioning, we must note oxytocin’s and vasopressin’s roles [70]. Vasopressin has a more notable effect on the male population, whereas oxytocin has a more notable effect on the female population. Animal studies report the regulatory role of estrogens in the production of OXT in the hypothalamus and the regulation of the number of OXTRs in the amygdala [25]. In males, however, a closer relationship has been observed between testosterone and its metabolite levels and vasopressin levels in the lateral septum. Furthermore, social recognition in males has been found to depend mainly on vasopressin V1a receptors [71,72]. Another issue that may result from the differences in the dominant sex hormones is related to CRH expression. In this regard, it has been reported that androgens inhibit CRH expression in the paraventricular nucleus of the hypothalamus, while estrogens stimulate it [73,74]. Of course, we cannot forget about the wide reference range for estrogen concentration in women. The same applies to the existence of many factors that significantly influence the concentration of this hormone, such as changes occurring during the menstrual cycle, pregnancy, and the postmenopausal period [75]. Perhaps it is also for these reasons that, in human studies, significant differences were observed within the female group.

Another intriguing issue, related to some extent to sex differences, concerns OXT receptor polymorphisms. Recent findings have reported that genetic variants concerning OXT receptor genes may cause changes in the structure and expression of the receptors. This can presumably influence patients’ responses to new treatment. Recently, rs535756 and rs2254298 gene polymorphisms have been of particular interest. Indeed, it seems to be one of the common alleles, rs53576A, that is associated with the promotion of deficits in social behavior related to attachment and empathy [76,77]. Apart from that, one meta-analysis found a relationship between this polymorphism and the general sociality of participants [78]. On the topic of sex-specific differences, literature reports indicate that sexual dimorphism determines the differences in the structure and functioning of the brain. This may depend on the occurring variant of the OXTR gene polymorphism. Regarding the polymorphism of rs2254298, the authors of the study drew similar conclusions. They investigated the role of that gene polymorphism in brain structure and function in default mode network (DMN)-related regions [79]. The DMN is a network of brain regions that show higher activity during rest, that is, periods that do not require significant cognitive resources or attention [80]. This network is also intriguing because it has recently become a point of interest for many scientists who associate some patterns of neural activity within this network with different mental disorders [81]. Other studies investigating the role of this common polymorphism have attended to the amygdala and its connectivity to other brain regions. In this case, too, the obtained results largely depended on the genotype [82]. However, considering the influence of genetic factors, we cannot ignore the phenomena related to epigenetic changes, that is, modifications of gene expression, which may happen throughout one’s life. The occurrence of this type of modification may be conditioned to some extent by lifestyle, including both diet and physical activity, as well as medications or stimulants taken. Exposure to stress factors is undoubtedly another factor contributing to epigenetic changes. Especially highly stressful events occurring in the early stages of life play a crucial role. The literature suggests that the coexistence of specific genetic variants within genes involved in neurotransmitter systems, combined with exposure to early life difficulties, may affect neurodevelopment [83]. This may occur mainly through epigenetic mechanisms, leading to changes in the structure, volume, and connectivity of key brain areas, such as the amygdala and the hippocampus. Consecutively, it could predispose to an increase in behavioral dysfunctions and mental health problems in the future and, consequently, contribute to obtaining divergent treatment results.

## 5. Further Directions

Unquestionably, in recent years, oxytocin has gained plenty of publicity in the scientific community. We know this based on the increasing number of publications regarding the use of this substance in various psychiatric diseases. This partially results from the good tolerability of the treatment and the numerous pleiotropic effects of oxytocin [84]. However, the studies conducted to date have had low statistical power. Additionally, the evidence is based mainly on animal and healthy population studies [85]. Hence, we do not know if these results can be extrapolated to the PTSD population. Another issue that hinders the interpretation of the studies is the dominance of studies that do not include women or gender differences in the results. This may contribute to unreliable results, owing to sexual dimorphism within the functioning of the oxytocinergic system. Furthermore, researchers have used different doses and frequencies of oxytocin administration. In the literature, we might find both single and multiple dosing. However, in the case of studies on the PTSD population, RCTs most often use single oxytocin doses of 24 IU or 40 IU [86]. The current research did not consider them to the extent that their impact on the later-obtained results could be determined. Therefore, we suggest considering patients’ blood cortisol concentrations in the future. We recommend such an approach because not all studies observed a decrease in the concentration of this hormone in the blood of patients with PTSD. Studies show mixed results, including both normal and increased cortisol levels. This could indicate the existence of several PTSD patient subpopulations, differing in slightly different disease courses. The disorder duration or even symptom severity could contribute to various outcomes, such as a longer or more frequent exposure to the inadequate activation of the HPA axis, which would lead to its subsequent dysregulation. Another factor worth considering in future research might be estrogen levels in the tested women, including the moment of the patient’s menstrual cycle if it occurs. Whenever possible, OXTR gene polymorphisms should also be measured. The inclusion of confounding variables would allow us to comprehend their influence on the obtained response more thoroughly, hence providing a more comprehensible understanding of PTSD pathophysiology. Thus, this would bring us closer to selecting patient groups that could benefit most from the proposed therapy.

Of course, a personalized approach to the patient does not involve focusing solely on the therapeutic potential of oxytocin. Tailoring treatment to specific patients would also have to apply to other therapeutics, for example, the substances acting on extinction memory, such as D-cycloserine, propranolol, and L-DOPA [87,88,89,90,91]. Another proposed agent associated with enhancing exposure therapy is methylene blue [92]. The effectiveness of adjuvant use has also been examined in the case of cognitive–behavioral therapy (CBT). In this respect, consideration was given mainly to methylenedioxymethamphetamine (MDMA). Phase III studies of MDMA-assisted psychotherapy (MAP) are ongoing, but the outcomes to date are promising [93]. Earlier, in 2017, the Food and Drug Administration (FDA) granted it breakthrough therapy designation (BTD) because of prior results. However, that type of psychotherapy assistance is targeted exclusively at treatment-resistant patients with PTSD. The MAP approach is also intriguing from the standpoint of our paper because of the potential role of oxytocin in its effects [94]. The effectiveness of this approach is seen mainly in influencing the memory of extinction. Nevertheless, the impact of the potential oxytocin release occurring after MDMA administration has not been ignored. Oxytocin may be responsible for pro-social and anxiolytic properties, and it may support the proper extinction memory function. Because MDMA mainly affects the serotoninergic system, it may be more beneficial to combine oxytocin treatment with SSRIs given the interdependencies between serotonin and oxytocin systems. We are aware that the effectiveness of MDMA-assisted psychotherapy also results from several therapeutic sessions, during which, due to a single strong MDMA effect on the serotonergic system, the patient can confront the event in a previously unattainable way. However, in our opinion, the lack of research verifying the potential benefits of the combination of SSRI and intranasal oxytocin therapy remains a missed opportunity, especially given the synaptic deficits in patients with PTSD and the potentially antidepressant role in promoting synaptic plasticity [95].

A personalized psychiatric approach would not be based solely on first-hand psychiatry knowledge or giving an unreasonably amount of time in the diagnostic office. Furthermore, by this, we do not mean the production of various questionnaires that objectify and improve the process of personalizing treatment. Modern technologies and the use of artificial intelligence seem to provide intriguing opportunities in this case. One of the studies in which researchers used machine learning can serve as an example here. In this study, artificial intelligence predicted which patient belongs to a particular subpopulation of patients with the use of appropriate algorithms. They were made based on the functional activity of the ION network, which is part of the DMN network [96]. Therefore, as shown in this example, we might also hold hopes for modern technologies. Although the current technologies are incapable of independently indicating the causes and effects of the observed specific deviations, due to the analysis and grouping of data, they can efficiently select similar groups of patients. Nevertheless, by knowing the effects of pathophysiological changes in a given subpopulation of patients, we will be able to determine the causes of their condition utilizing backward reasoning with the help of artificial intelligence, and this, in turn, would allow for better future potential therapeutic options to be targeted, focusing on those who can obtain the best results from them. In other words, we would target the substance directly to the areas that require intervention in a given patient. This would bring us closer to personalized medicine, which is more marked in such branches of medicine as oncology and hematology. In these fields, due to the possession of specific markers, clinicians can propose optimal therapeutic solutions to some of their patients [97].

To sum up, in our opinion, it might be feasible to select appropriate, objective markers that enable the division of patients with PTSD into individual subpopulations that are similar in terms of the disease picture. These markers would include both specific hormonal profiles, which provide us with information about the nature of the pathophysiological changes, and specific patterns of neuronal activity. Thanks to them, there would be the possibility to find the best pharmacological treatment for patients, making it possible to remove the cause of the disease and not simply mainly eliminate the symptoms occurring in patients.

## 6. Conclusions

Despite the lack of a comprehensive understanding of oxytocin’s mechanism and the low number of conducted studies, the administration of oxytocin seems to be a promising therapeutic strategy for patients with PTSD. This is due to the increasing number of reports on its pleiotropic effects related to the pathophysiologically changed areas in this population. The nasal spray form of oxytocin administration can affect crucial brain structures from a mental health perspective. This is possible due to the dense distribution of oxytocin receptors and the projections of the oxytocinergic system to several brain structures, such as the paraventricular nucleus of the hypothalamus, the raphe nuclei, the amygdala, the hippocampus, and the prefrontal cortex. Bearing that in mind, the reduction in depressive and anxiety symptoms, the modification of the sympathetic system and HPA axis activity, and the effect on the consolidation of memory pathways (including fear extinction) are not surprising. Furthermore, these properties might influence the development or course of some common comorbid disorders in this population, primarily sleep disorders, autoimmune and cardiovascular diseases, and metabolic disorders. However, it should be noted that the oxytocinergic system has high plasticity, and, consequently, the therapeutic results of this therapy may depend on many confounding factors. Hence, we suggest considering some of the above-mentioned issues, such as the patient’s gender and hormonal profile, and the polymorphism of the oxytocin receptor, in future studies.

To sum up, the potential properties of oxytocin may facilitate the process of PTSD treatment mainly by modulating the course of the stress reaction, supporting extinction memory, and increasing involvement in therapeutic activities. They might also contribute to a reduction in the intensity of comorbid disorders and their progression, or they might even prevent their development. However, without understanding the relationship between inter-individual variables and the observed response to treatment, we will not be able to identify the subpopulation of patients who would benefit most from this type of treatment.

## Figures and Tables

**Figure 1 jpm-12-01067-f001:**
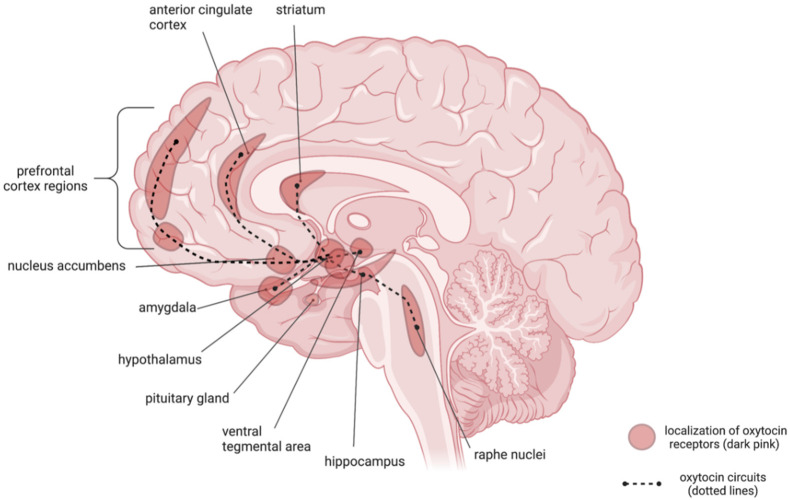
Localization of oxytocin receptors (dark pink) and oxytocin circuits (dotted lines) (created with BioRender.com, accessed on 18 June 2022).

**Figure 2 jpm-12-01067-f002:**
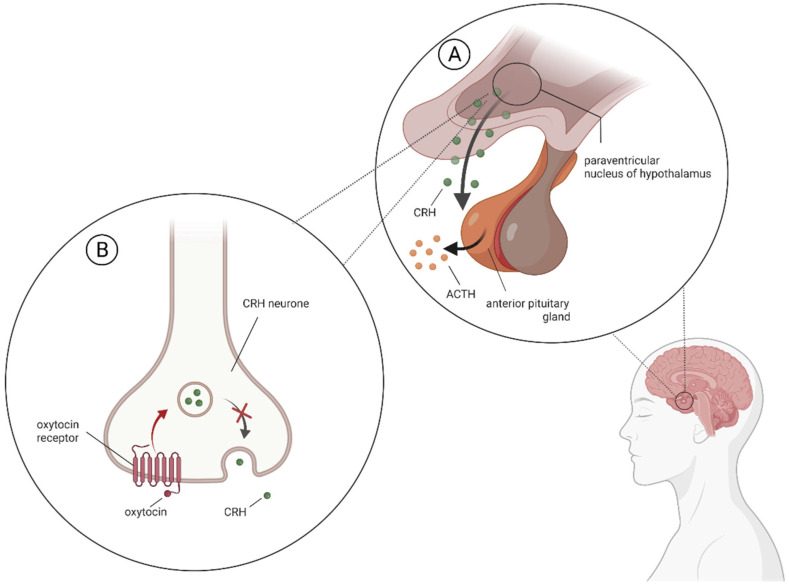
The functioning of the hypothalamic–pituitary–adrenal (HPA) axis in the brain with the oxytocin effect on CRH neurons (created with BioRender.com, accessed on 18 June 2022). (**A**) After a stressor, corticoliberin (CRH—corticotropin-releasing hormone) is released from the paraventricular nucleus (PVN) of the hypothalamus. It acts on the anterior pituitary gland, thereby mediating the release of corticotropin (ACTH—adrenocorticotropic hormone). This hormone leads to the release of glucocorticoids, among others, from the adrenal cortex. (**B**) OXT binds to OXT receptors, located on CRH neurons within the hypothalamus. As a result, there is a reduction in CRH secretion. As a further consequence, components of the HPA axis are affected, potentially contributing to modification of the pathophysiologically dysregulated axis among patients with PTSD.

**Figure 3 jpm-12-01067-f003:**
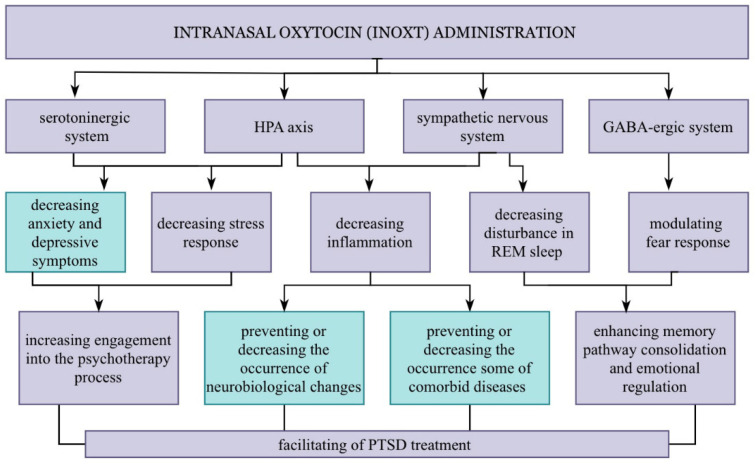
Theoretical relationship between using INOXT and relevant aspects from a PTSD point of view. Intranasal oxytocin (INOXT) administration can induce several potentially beneficial effects for PTSD patients (purple ones) by affecting brain structures. Its properties might facilitate PTSD treatment mainly by affecting memory, including fear extinction memory, and increasing engagement in the psychotherapy process. Apart from that, oxytocin administration might prevent or decrease the occurrence of neurobiological changes or some comorbid symptoms and diseases (teal ones).

## Data Availability

Not applicable.

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
