# Peer review of "Progress in Personalized Psychiatric Therapy with the Example of Using Intranasal Oxytocin in PTSD Treatment"

_jpm, 2022, doi:10.3390/jpm12071067_

Round 1

Reviewer 1 Report

I read with interest the present article on “Progress in personalized psychiatric therapy on the example of using intranasal oxytocin in PTSD treatment” for the Special Issue "Personalized Treatment and Diagnosis Strategies in Psychiatry".

The article is well written, the aim is interesting however the study design is unclear.

This is a narrative review that aims “to identify the main factors responsible for different therapeutic responses” however, there is no methodological part that accurately describes how the authors selected the articles taken into consideration. Although it is not a systematic review, authors should better describe what kind of criteria they used to select articles.

Furthermore, the authors affirm that “in the PubMed database phrases “PTSD” and “oxytocin” we get 27 results of randomized controlled trials (RCTs) carried out on that population” but later it is not clear if all RCTs were considered and if not, what were the exclusion or inclusion criteria since it seems that not all of them are mentioned.

A table or a flow chart could help to understand better methodological design of the review.

Reviewer 2 Report

JPM-1758385 v1 peer review. England, June 7th, 2022

The review article by Szafoni and Piegza is investigating personalized medicine in the context of PTSD treatment, using intranasal oxytocin. The article is easy to follow and the authors have included a few figures, which will be greatly appreciated by readers. The quality of these figures is excellent, and authors can be commended for that (although small alterations are needed, please see below). At present, the manuscript requires a few minor alterations, highlighted point-by-point below. The authors should consider addressing these to improve the quality of their manuscript. Thus, I recommend minor revisions. I hope the authors will find my comments useful.

 Throughout: authors should place references before the full stop. Authors should revise the grammar/English.

 Throughout: authors are using OXT as the abbreviation for Oxytocin. Thus, intranasal oxytocin should be abbreviated as “INOXT” instead of “INOT”. Please edit throughout the manuscript and in the Figures.

 Abstract, line 16. The authors should find a more suitable adjective (“conscientiously”).

 Section 1, line 23. When authors mention that “medical records and literary works testify to the fact”, references should be cited.

 Section 1, line 44. Authors should use inclusive language. Please avoid using “PTSD population”. Suggestion: “in patients suffering from PTSD”.

 Section 1, lines 75-77. Here, authors should avoid given so little details. Indeed, the Pubmed database is not the only database that should be used for searching clinical trials. Since authors do not aim to conduct a meta-analysis in their current manuscript, they should either remove this information or should check all available databases. For the latter, authors are invited to check a very good example of this, published in JPM very recently, at the following: https://doi.org/10.3390/jpm12040651. Furthermore, authors have cited a good meta-analysis using OXT and imaging, referenced as reference number 44 (Wang, Yan, Li and Ma). These authors have provided a rather good methodology for finding clinical studies (Embase, etc…). However, the full reference, displayed on lines 506-507 does not include the Journal’s name. Please insert Social Cognitive and Affective Neuroscience in the bibliography.

 Section 2, line 83. Authors should indicate the full name of “CAPS”.

 Section 2, line 83, odd spacing between “in” and “participants”. Please edit.

 Title of section 3, line 91. “Pleiotropic” instead of “Plejotropic”. Please edit.

 Legend of Figure 1, lines 100-101. Here, the legend should be rephrased. At present, it suggests that receptors possess projections. Suggestion: “Localization of oxytocin receptors (dark pink) and oxytocin circuits (dotted lines).” Please also edit the key directly on display.

 Legend of Figure 2, lines 142-144. What is the purpose of these? “This is a figure”, “Schemes follow another format”, “if there are multiple panels, they should…”. Please edit.

 Legend of Figure 2, lines 148-149. Authors should rephrase the beginning of this sentence. At present, it suggests that OXT binds to its receptor only when administered (“after administration”). Suggestion: start directly the sentence with “OXT binds to OXT receptors, located on CRH neurons within the hypothalamus”.

 Legend of Figure 3, lines 205-225. Please provide a more succinct legend. Furthermore, what are the differences between purple and green boxes? Authors provided a legend for “blue ones”, but what about the purple ones? Also, “blue” might not be the best description of the chosen colours, which appears to be more in green shades (teal/cyan). Please edit.

 Section 4, line 325. “L-DOPa”. Please edit.

 Section 5, line 379. Please refrain from criticizing published (and thus peer-reviewed) studies using comments such as “low statistical power”. Previously-published studies report either significant or non-significant effects. Do authors rather mean “a low number of studies”? Please edit.

Round 2

Reviewer 1 Report

The article can now be accepted for publication